# Nanostructured Layer of Silver for Detection of Small Biomolecules in Surface-Assisted Laser Desorption Ionization Mass Spectrometry

**DOI:** 10.3390/ma15124076

**Published:** 2022-06-08

**Authors:** Adrian Arendowski, Gulyaim Sagandykova, Radik Mametov, Katarzyna Rafińska, Oleksandra Pryshchepa, Paweł Pomastowski

**Affiliations:** 1Centre for Modern Interdisciplinary Technologies, Nicolaus Copernicus University in Toruń, Wileńska 4, 87-100 Toruń, Poland; adrian@arendowski.hub.pl (A.A.); mametov.radik@gmail.com (R.M.); pryshchepa.alexie@gmail.com (O.P.); pomastowski.pawel@gmail.com (P.P.); 2Department of Environmental Chemistry and Bioanalytics, Faculty of Chemistry, Nicolaus Copernicus University in Toruń, Gagarina 7, 87-100 Toruń, Poland; katrafinska@gmail.com

**Keywords:** mass spectrometry, electrodeposition, silver layer, laser desorption/ionization, small biomolecules

## Abstract

A facile approach for the synthesis of a silver nanostructured layer for application in surface-assisted laser desorption/ionization mass spectrometry of low-molecular-weight biomolecules was developed using electrochemical deposition. The deposition was carried out using the following silver salts: trifluoroacetate, acetate and nitrate, varying the voltage and time. The plate based on trifluoroacetate at 10 V for 15 min showed intense SALDI-MS responses for standards of various classes of compounds: fatty acids, cyclitols, saccharides and lipids at a concentration of 1 nmol/spot, with values of the signal-to-noise ratio ≥50. The values of the limit of detection were 0.71 µM for adonitol, 2.08 µM for glucose and 0.39 µM for palmitic acid per spot. SEM analysis of the plate showed anisotropic flower-like microstructures with nanostructures on their surface. The reduced chemical background in the low-mass region can probably be explained by the absence of stabilizers and reducing agents during the synthesis. The plate synthesized with the developed approach showed potential for future use in the analysis of low-molecular-weight compounds of biological relevance. The absence of the need for the utilization of sophisticated equipment and the synthesis time (10 min) may benefit large-scale applications of the layer for the detection of various types of small biomolecules.

## 1. Introduction

The analysis of low-molecular-weight (LMW) compounds has significance in various areas of research, such as forensics [1], the medicine and pharmaceutical industry [2], food safety [3,4] and environmental studies [5]. In some studies, the research on small molecules focused mostly on their quantification and distribution in various matrices. Such an applied character of the analysis of LMW compounds can be complemented by the collection of molecular profiles of various biological samples for screening purposes. Coupling data on molecular profiles to advanced data processing algorithms facilitates the development of platforms for various goals. Analysis and comparison of the levels of small biomolecules in the organism can be useful in studies of metabolomics [6], the search for cancer biomarkers [7], disease pathogenesis [8], differentiation between stages [9], reasonable prognosis and the careful selection of the treatment strategy [10]. 

In turn, small molecules can be successfully analyzed using a type of laser mass spectrometry, which is surface-assisted laser desorption/ionization (SALDI). According to the IUPAC definition, SALDI is a group of matrix-free methods used in laser desorption/ionization MS analysis of biologically derived molecules [11]. In SALDI-MS, a surface with various nano- and microstructures, comprised of crystallites of nanoparticles, assists the ionization and desorption of the analytes [12]. For the first time, surface-assisted laser desorption/ionization was reported in a 1995 article by Sunner et al. [13]. The active system used in the research consisted of graphite microparticles of 2–150 μm suspended in glycerol. The MS spectra obtained by the authors were characterized by the presence of a low chemical background; therefore, similar methods based on nanoparticles were successfully used in the analysis of LMW compounds. Over the last few years, many different types of surfaces have been developed for use in SALDI MS. These surfaces were developed based on nanoparticles of metals [14], metal oxides [15], carbon [3] and silicon [16]. 

Great endeavors have been undertaken to develop methods for the preparation of nanomaterials that have been used in the fabrication of SALDI MS targets for the analysis of small molecules of biological relevance. *Zhang* et al. reported Multiplexed Nanomaterial-Assisted LDI for Cancer Identification (NALCI) for cancer diagnosis and classification [17]. NALCI was applied in 1183 individuals, including 233 healthy controls and 950 patients diagnosed with lung, liver, pancreatic, thyroid, gastric and colorectal cancers. The results showed 93% sensitivity and 91% specificity in the differentiation between cancer and controls in the internal validation group compared to 84% sensitivity and specificity in the external validation group, and up to eight biomarkers were distinguished. Kołodziej et al. reported the application of silver-109 nanoparticles for the detection of carboxylic acids such as azelaic, 3-methylhippuric, linoleic, oleic, arachidic and erucic acids [18]. ^109^AgNPs were prepared using pulsed fiber laser (PLF) 2D GS (galvano-scanner) LASiS (laser ablation synthesis in solution) and were applied for the detection of carboxylic acids in spiked human blood serum to assess the matrix effect and semi-automatic mass spectrometry imaging [18]. Mesoporous PdPtAu spheres were utilized to reveal the specific metabolic phenotype of gastric cancer and resulted in precise diagnoses with an area under the curve of 0.942 [19]. Pd-Au synthetic alloy with a core–shell structure supported with magnetic nanoparticles allowed for the differentiation of medulloblastoma patients from healthy controls with a sensitivity of 94.0%, a specificity of 85.7% and an accuracy of 89.9% using machine learning. In addition, the process of radiotherapy in patients was monitored, and a preliminary panel of metabolites in serum was identified [20]. Mizoshita et al. designed nanoporous substrates with molecular-level perfluoroalkyl/alkylamide surface for LDI-MS of small proteins [21]. Surface functionality was tuned by mixing perfluoroalkyl/alkylamide groups at a molecular level, and the amyloid fragment, which was suggested as a biomarker of Alzheimer’s disease, was detected at a concentration of 0.2 fmol/µL in the presence of other blood compounds [21]. Palladium nanoparticle–decorated thiol-functionalized metal–organic framework (MOF) nanocomposite was fabricated for the sensitive analysis of oligosaccharides [22]. 

Unlike both wet and dry chemical methods, electrodeposition/electroplating techniques for the preparation of nanostructured layers do not require the use of reducing reagents and stabilizers. The use of reducers and stabilizers can result in a rich chemical background, complicating spectra interpretation. Furthermore, in situ synthesis of the nanostructured layer may be an advantage for the minimization of “hot spots”. In contrast, the deposition method, which does not require sophisticated equipment for chemically synthesized nanoparticles, is a so-called “dried-droplet method” [23]. Such deposition may lead to the occurrence of a coffee-ring effect that results in the heterogeneity of the sample spots and low shot-to-shot reproducibility [24]. 

The use of silver nanostructures as active surfaces in SALDI-MS enables excellent absorption of UV radiation emitted by lasers in spectrometers and efficient energy transfer, ensuring the desorption of analyte molecules. The advantages of silver nanostructures in bioanalysis include relatively high tolerance to the presence of salt and the possibility of using internal calibration with the signals of silver clusters [25]. In addition, hot electrons, which play a crucial role in the LDI process, in plasmonic noble metal nanostructures can serve as a source of charges assisting ionization [26]. The design of plasmonic metal nanostructures with a high yield of hot electrons can lead to enhanced LDI efficiency [19]. 

Electrodeposition techniques for the synthesis of silver nanostructures have also been reported by several authors [27,28] but have not received much attention compared to chemical methods. The lack of literature on the effects of the parameters of electrodeposition on SALDI efficiency represents a research gap in this area. 

In this study, a new approach for the synthesis of the SALDI-MS plate was developed using electrodeposition. The synthesis was carried out with the reduction of various silver salts directly on the target/electrode at various values of voltage and time. The silver layer synthesized from silver trifluoroacetate at 10 V for 15 min was selected as sensitive to various classes of small biomolecules. Anisotropic flower-like microstructures with nanostructures on their surfaces were observed with a scanning electron microscopic analysis of the obtained layer. Moreover, the application of the reported range of the synthesis voltage (3–30 V) for silver nanostructured materials has not been studied in terms of LDI-MS efficiency or reported elsewhere. 

## 2. Materials and Methods

### 2.1. Reagents and Materials

All standards used to study the SALDI performance of synthesized plates were of the highest commercially available purity: adonitol, D-glucose, cholesterol, palmitic acid, caffeine, linolenic acid and *myo*-inositol (Sigma-Aldrich, Steinheim, Germany). In addition, all solvents were of LC-MS purity (water, iso-propanol, acetonitrile, acetone and chloroform) and purchased from Sigma-Aldrich (Steinheim, Germany). Precursors for synthesis: silver trifluoroacetate (AgTFA, 98%), acetate (AgAc, 99%) and nitrate (AgNO_3_, 99%) were purchased from Trimen Chemicals (Łódż, Poland). 

### 2.2. Sample Preparation

The stock solutions of adonitol, caffeine and glucose were prepared in 1.5 mL Eppendorf tubes by dissolving in water, while stock solutions for cholesterol and palmitic and linolenic acids were dissolved in chloroform using 1.5 mL glass vials and manual glass syringes for sample preparation (Agilent, Santa Clara, CA, USA). Representatives of several classes of compounds such as adonitol, glucose, cholesterol and palmitic acid at a concentration of 1 mg/mL were used as test compounds for the selection of the synthesis parameters. Spectra from the final plate were collected using freshly prepared solutions of adonitol, *myo*-inositol, caffeine, glucose, cholesterol, and palmitic and linolenic acids at a concentration of 1 nmol/spot. Solutions of adonitol, glucose and palmitic acid at concentrations of 0.5, 1, 2, 5 and 10 µg/mL for the determination of LLOQ were prepared by dilution of the stock solutions in water (adonitol, glucose) and chloroform. 

### 2.3. Synthesis of SALDI Plates 

All plates were prepared by cutting a sheet of stainless steel (H17) to a size of 2.5 × 7.5 × 0.8 cm. Cleaning of the plates was carried out by sonication in the following solvents: acetone, methanol and acetonitrile for 10 min for each solvent in the metal box. Each plate was additionally rinsed with iso-propanol prior to synthesis. Electrodeposition was carried out using Consort EV202 (Turnhout, Belgium) power supply for electrophoresis. Two pieces of steel were placed vertically parallel to each other using a 100 mL glass beaker and connected to the power supplier. The plate for surface-assisted laser desorption ionization was connected to the negative slot (negative electrode), and an additional electrode was connected to the positive slot (positive electrode). The synthesis was carried out using different voltage values (3, 10 and 30 V) and times (10, 15 and 20 min) using the following precursors: silver trifluoroacetate, acetate and nitrate. The volume of the solution of the precursor was 90.0 ± 0.5 mL, and all precursors were used at the same molar ratio (9 × 10^−5^ mol). Further experiments for the evaluation of SALDI performance were carried out by synthesis using AgTFA as a precursor at 10 V and for 15 min. The distance between the plates was 3.6 cm. The solvent used for the reaction was a mixture of acetonitrile (20%) and isopropanol (80%). After each synthesis, each plate was washed with a cotton ball and boiling isopropanol to remove aggregates. 

### 2.4. SALDI TOF-MS Analysis

Analysis was carried out in positive reflector mode in the mass range *m/z* 80–2000. All spectra were collected using the ultraFlextreme MALDI-TOF-MS instrument (Bruker Daltonics, Bremen, Germany) equipped with a modified neodymium-doped yttrium aluminum garnet (Nd:YAG) laser operating at 355 nm and a frequency of 2 kHz. Data processing was performed using FlexAnalysis 3.3 software (Bruker Daltonics, Bremen, Germany) and mMass 5.5.0. [29]. A 0.5 µL sample was spotted onto the synthesized plate inserted in the holder for ITO glass slides for MALDI imaging (Bruker Daltonics, Bremen, Germany). Mass calibration was carried out with a cubic enhanced calibration strategy using the signals of silver. The number of laser shots was 1500 (for the first experiments) and 2000 (for final experiments for SALDI efficiency). Reflector voltages were 26.64 and 13.54 kV, while the first accelerating voltage was 25.08 kV, and the value for the second ion source voltage was 22.43 kV. The value of detector gain for the reflector was 30×. The value of the global attenuator offset was 42%. Theoretical *m/z* values of analyzed compounds were confirmed by using the ChemCalc program [30]. Signals selected for putative identification had S/N values ≥10, and propositions for metabolites were based on the mass error, which was |**Δ|** ≤ 5 ppm. Heatmaps were prepared using the GraphPad Prism software (version 8.0.1., San Diego, CA, USA). Calibration curves for the determination of LOD and LLOQ were prepared using the S/N value as a function of analyte concentration, similar to reports in other publications [27,31]. The LOD and LLOQ parameters were determined using the signals corresponding to adducts with the Ag-107 isotope, which were the most intense among all adducts. 

### 2.5. SEM Analysis

The plate synthesized using AgTFA precursor at 10 V for 15 min was attached to the SEM table with carbon tape and subjected to scanning electron microscopy (SEM) analysis. SEM images were obtained using SEM/FIB instrument (Quanta 3D FEG FEI, Gräfelfing, Germany) operating at 20 kV and in SE mode.

## 3. Results and Discussion

### 3.1. Effects of Precursor, Applied Voltage and Synthesis Time on SALDI Efficiency of Low-Molecular-Weight Compounds 

Since the precursor, voltage and time of the reaction play crucial roles in electrochemical deposition, the effects of those parameters on the SALDI performance of the resulting plates were studied. Moreover, the literature review of previous publications on the electrochemical deposition of SALDI substrates showed that authors applied voltages <3 V for longer times using silver nitrate [28] and chloride salts [27]. Therefore, the following ranges were selected to study the effects of voltage and time: 0–30 V and 10–20 min, respectively. 

First, the synthesis of SALDI plates was carried out at 10 V for 15 min as conditions in the middle of the tested ranges using different precursors. One representative of each of the various classes of low-molecular-weight compounds of biological significance, such as cyclitols (adonitol), saccharides (glucose), fatty acids (palmitic acid) and lipids (cholesterol), at a concentration of 1 mg/mL was selected to evaluate the SALDI-MS efficiency of each plate. The responses of all analytes (Figure 1) for the plate based on silver trifluoroacetate (plate 3) were higher as compared to those based on silver nitrate (plate 6) and silver acetate (plate 7). Such differences can probably be explained by the solubility of silver salts. Therefore, further study of the effect of the synthesis time and voltage was continued using plates synthesized with the utilization of silver trifluoroacetate as a precursor. 

For this purpose, the synthesis was carried out using silver trifluoroacetate as a precursor using the following conditions: 3 V (plate 1), 10 V (plate 3) and 30 V (plate 5) with 15 min as a controlled variable. The value of 15 min was selected since it was in the middle of the tested range. The results showed that plate 3 provided the highest responses for all analytes except adonitol (Figure 1). However, the SALDI MS analysis of the plates synthesized at 10 V and 30 V showed a higher number of additional signals in the range <*m/z* 500 for plate 5 that may potentially cause interferences (Figure 2). The observed signals can potentially be explained by side reactions in the solution during the deposition at higher values of voltage.

The next parameter of the synthesis was time. Since this effect has not been studied in previous publications, consideration of this parameter was attempted in our study. The plates were synthesized and evaluated for SALDI efficiency using silver trifluoroacetate with 10 V as a controlled variable and the following values of time: 10 min (plate 2), 15 min (plate 3) and 20 min (plate 4). According to the obtained results (Figure 1), the responses for plate 3 were found to be the highest for glucose and palmitic acid, while the responses of cholesterol were similar for plates 2 and 3. In contrast, responses for adonitol were the highest for plate 2; however, the differences between plates 2 and 3 accounted for 23% for both ^107^Ag and ^109^Ag adducts. Since SALDI efficiency may differ for various analytes, it was attempted to select a plate that was efficient for different classes of analytes for potential utilization in the analysis of small biomolecules in samples with complex compositions. Thus, one more parameter, namely, the S/N ratio, was evaluated. The values of the S/N ratio for adonitol amounted to 82.8/76.1 (^107^Ag/^109^Ag) for plate 3 vs. 129.6/120.6 (^107^Ag/^109^Ag) for plate 2. For cholesterol, such values were found to be 162.6/169.1 (^107^Ag/^109^Ag) for plate 3 and 277.2/249.9 (^107^Ag/^109^Ag) for plate 2. The similarities in the values of S/N allowed us to suggest that both plates 2 and 3 can be considered suitable for adonitol and cholesterol. 

Therefore, plate 3 (10 V, 15 min, silver trifluoroacetate) was selected as sensitive to various classes of analytes and was used for the further study of SALDI efficiency for low-molecular-weight compounds. 

### 3.2. SEM Analysis 

SEM analysis of the plate at an accelerating voltage of 20 kV showed the formation of a layer (Figure 3A–C) composed of flower-like structures in the size range between 0.5 and 1 μm (Figure 3D). An interesting feature of the obtained material is the presence of nanostructures on the surface of microstructures (Figure 3E,F). The presence of nanostructures on the microstructures without a phase boundary between them could affect the interactions of the layer with light and analytes.

In addition, deposited microstructures demonstrated nearly homogeneous distribution on the surface (Figure 3A–C). The SEM picture of the plate used as a positive electrode (Appendix A) showed an insignificant number of nanoparticles, which were observed as single units with an approximate size of 20–50 nm. It is suggested that those nanoparticles did not grow into microstructures due to the opposite electrode charge and low silver cation migration rates. Signals of silver (Ag^+^ and Ag_2_^+^) with low intensity were also observed in SALDI MS spectra of the plate used as a positive electrode (Appendix A).

### 3.3. SALDI MS Performance

SALDI MS performance of the plate with the nanostructured silver layer was evaluated using standards of low-molecular-weight compounds at a concentration of 1 nmol/spot. According to the obtained results (Figure 4), intense signals of adducts of analytes with potassium [M+K]^+^, sodium [M+Na]^+^ and silver [M+^107^Ag]^+^/[M+^109^Ag]^+^ were observed in the spectra with S/N values ≥50. The values of S/N potentially show that the limit of detection for the studied small molecules was less than 1 nmol. The adducts of analytes with sodium and potassium were expected since trace amounts of them can be present in water and solvents used for the synthesis, even after purification.

In an attempt to assess the potential mechanism of ionization for the tested analytes, values of the intensity and S/N ratio were compared (Figure 4). Adducts of analytes with silver [M+Ag]^+^ for both isotopes were the most intense for all analytes. This can indicate that ion formation was preferably assisted with silver cations. These data are similar to the results reported by Cioffi et al. [28], who also utilized the electrochemical method for the synthesis of the SALDI plate, and ion formation was assisted mainly by the cationization of analytes with silver ions. The low intensities of signals corresponding to [M+H]^+^ (Figure 4) potentially indicate that the ionization of all of the studied analytes via hydrogen adduct generation was not preferable. The signals corresponding to [M+^107^Ag/^109^Ag]^+^ showed comparable intensities of >3 × 10^5^ a.u., except for caffeine, which has the most distinct structure among the analytes, possessing nitrogen atoms in the purine ring. Notably, the formation of silver adducts [M+Ag]^+^ is often used as an analytical strategy to stabilize nonpolar analytes and decrease their limit of detection [32].

Intense signals of adducts of palmitic acid with silver [M+^107^Ag/^109^Ag]^+^ and sodium [M+Na]^+^ were observed. In contrast, linolenic acid showed a low-intensity signal of the [M+Na]^+^ adduct, which probably correlates with the number of double bonds in the structure since linolenic acid has three double bonds in the structure as compared to the saturated linear structure of palmitic acid.

The data showed that the layer was sensitive to various small biomolecules that differ in chemical structure and physico-chemical properties (Figure 4).

### 3.4. Determination of the Limit of Detection and Quantification

The following compounds were selected for the determination of the limit of detection (LOD) and the lower limit of quantification (LLOQ): adonitol, glucose and palmitic acid at concentrations of 0.5, 1, 2, 5 and 10 µg/mL. The LOD and LLOQ values are shown in Table 1, and the intensity of the signals for each analyte and the linear regressions based on the signal-to-noise (S/N) ratio are shown in Appendix A.

The plotted linear regressions showed a good fit for glucose, palmitic acid and adonitol (Table 1). The limit of detection for adonitol was 107.95 ng/mL or 0.71 μM, corresponding to 53.98 pg/spot or 0.35 pmol/spot. Adonitol showed a value of RSD of 2.16%, which is an excellent result since the value is <5%. Unlike adonitol, glucose and palmitic acid showed higher values. However, these high values were obtained for concentrations, which were determined as the limit of detection. The value of RSD for glucose was 11.08% for a concentration of 1 μg/mL, which is an adequate value since it is close to 10%. Since there are no publications reporting LDI techniques for the detection of polyols, including adonitol, the approach reported in this study may find applications in the screening of biological fluids since it enables sensitive analysis with 0.5 μL of sample volume and fast data acquisition. For example, Lee et al. [33] reported a GC-MS technique with a LOD value for adonitol of 1 μg/mL, which is much higher, potentially indicating that even with consideration of the effect of the matrix, the nanostructured silver layer should be suitable for the detection of adonitol for screening purposes. Another potential application may be found in the assessment of cyclitols isolated from plants. Since cyclitols possess plenty of beneficial properties for human health [34], new extraction techniques have been developed [35] to reduce the costs of the potential production of bioactive supplements and assessment of the nutritional quality of the food. Furthermore, it is crucial to study the metabolism of cyclitols for the determination of their biological activity for the pharmaceutical industry, and sensitive analytical techniques can assist in the determination of the intermediates/products of metabolic pathways. For example, polyol pathways facilitate the accumulation of polyols in aldose-reductase deficiencies under hyperglycemic conditions, leading to the early damage of the tissue and inborn error [36].

The LOD for glucose was 374.18 ng/mL or 2.08 μM, corresponding to 187.09 pg or 1.04 pmol per spot. In the case of chromatographic techniques, it is possible to detect glucose at lower concentrations [37]; however, the need to separate the complex matrix of the sample remains challenging. Silina et al. [27] reported the SALDI method based on silver nanoparticles synthesized using the electrochemical technique, which allowed them to obtain a LOD value of 7 ng/spot, which is a higher value as compared to the one reported in the current study (Table 1). Such differences in the LOD value can be explained by differences in the morphologies of the nanostructured substrates and, subsequently, interactions of the analyte, thus affecting the ionization efficiency. Furthermore, substrate morphology also determines the optical properties of the material, also affecting the ionization process. Besides food research, glucose determination has found applications in the studies of glucose metabolism in medical research [38,39]. The RSD value for glucose can be potentially improved in the future with an automated data collection procedure with an increased number of shots, while the manual procedure that was utilized in the current study was necessary for the stage of characterization of LDI performance.

The LOD for palmitic acid was 98.76 ng/mL or 0.39 μM, corresponding to 49.38 pg or 0.19 pmol per spot. Using nanostructure-initiator mass method spectrometry (NIMS) as an LDI technique, Reindl et al. [31] achieved a LOD value for palmitic acid at a level of 0.1 pmol/spot, which is almost two times lower than the value obtained for the silver nanostructured layer (Table 1). Kołodziej et al. [40] reported the limit of detection at 50.9 µM or 13,056 ng/mL using the SALDI technique with the application of chemically synthesized silver nanoparticles. This may indicate that the proposed nanostructured silver layer can be considered one of the most sensitive LDI techniques for the detection of palmitic acid.

Finally, the obtained LOD values potentially demonstrate that the reported approach allows for the sensitive determination of both water-soluble and water-insoluble compounds.

## 4. Conclusions

In conclusion, we propose an approach for the synthesis of the silver layer for SALDI MS targets based on electrodeposition directly on the target. The effects of the precursor, time and voltage on ionization efficiency were studied using standards of low-molecular-weight compounds of biological significance. Based on the obtained results, the plate with a layer reduced from silver trifluoroacetate at 10 V for 15 min was selected for further analysis of standards of LMW at a concentration of 1 nmol/spot. In turn, the SEM analysis of the obtained layer showed flower-like structures with the specific feature that nanostructures were included on the surface of microparticles. Finally, the proposed approach allowed us to obtain the values of the limit of detection for selected low-molecular-weight compounds at the level of ng/mL, with values of relative standard deviation <20%.

Thus, the plate demonstrated a low chemical background and showed sensitivity to various classes of small biomolecules. The proposed synthesis is relatively fast and can be accomplished without the need to utilize sophisticated equipment that is unavailable in small laboratories. Finally, the plate showed prospects for applications in SALDI MS analysis of low-molecular-weight compounds of biological relevance.

## Figures and Tables

**Figure 1 materials-15-04076-f001:**
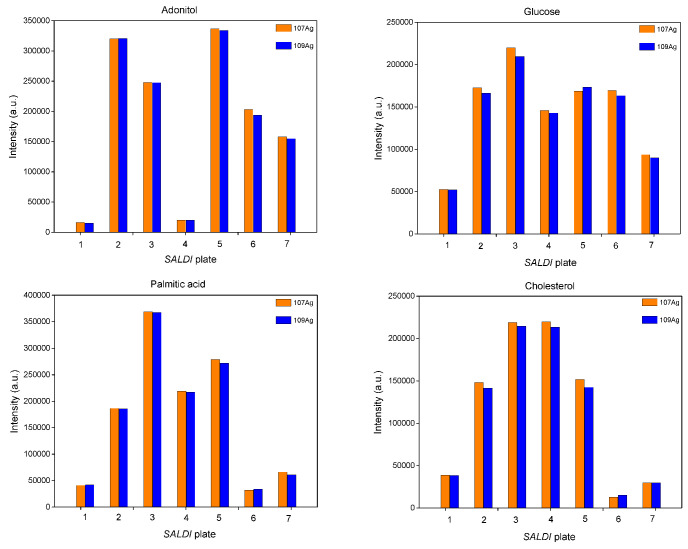
Effects of precursor, applied voltage and time on SALDI efficiency of adonitol, glucose, palmitic acid and cholesterol as representatives of different classes of compounds; SALDI plates: 1—AgTFA, 3 V, 15 min; 2—AgTFA, 10 V, 10 min; 3—AgTFA, 10 V, 15 min; 4—AgTFA, 10 V, 20 min; 5—AgTFA, 30 V, 15 min; 6—AgNO_3_, 10 V, 15 min; 7—AgAc, 10 V, 15 min.

**Figure 2 materials-15-04076-f002:**
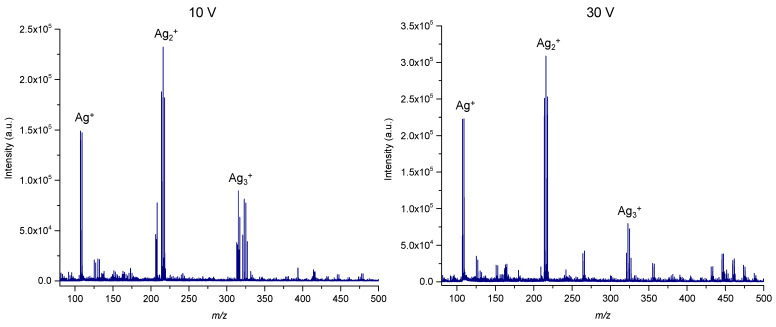
SALDI MS analysis of the plates synthesized with 10 and 30 V and 15 min as a controlled variable.

**Figure 3 materials-15-04076-f003:**
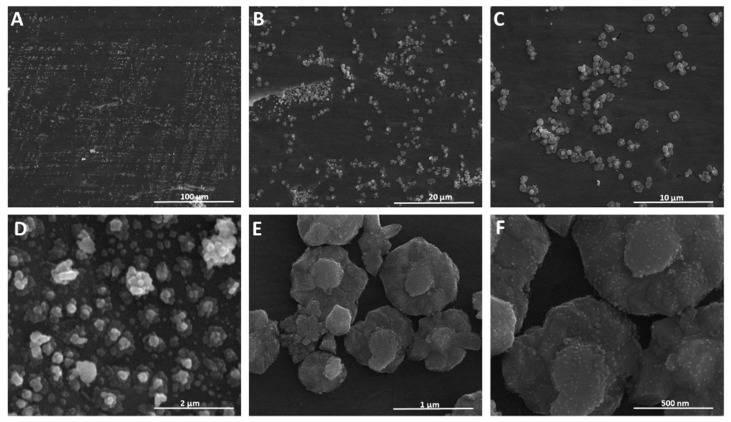
SEM images of SALDI plate synthesized with AgTFA as a precursor (10 V, 15 min) at different values of magnification, (**A**)—1000×; (**B**)—5000×; (**C**)—10,000×; (**D**)—50,000×; (**E**)—100,000×; (**F**)—200,000×.

**Figure 4 materials-15-04076-f004:**
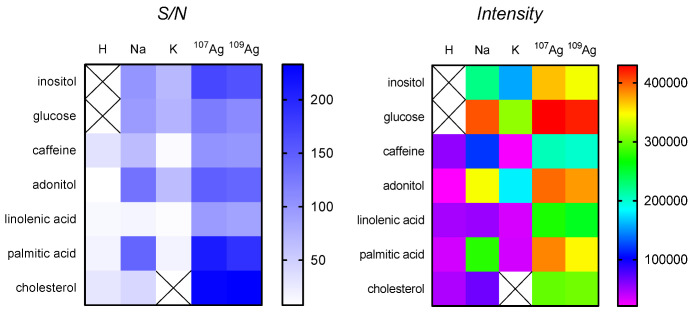
Heatmaps of intensities and S/N values for adducts of LMW compounds with hydrogen, sodium, potassium and silver isotopes plotted from SALDI MS spectra of standard solutions at a concentration of 1 nmol/spot and 2000 shots.

**Table 1 materials-15-04076-t001:** Parameters of regression equations, limit of detection and lower limit of quantification for studied compounds.

Analyte	LOD ^a^ [ng/mL]	LLOQ ^b^ [ng/mL]	RSD ^c^ [%]	Regression Equation	r^2^
Adonitol	107.95	179.91	2.16	y = 9.073x + 8.5833	0.998
Glucose	374.18	623.63	19.02	y = 5.0072x + 7.0728	0.968
Palmitic acid	98.76	164.60	14.84	y = 7.3103x + 18.045	0.980

^a^ Based on an S/N ratio of 3; ^b^ based on an S/N ratio of 5; ^c^ relative standard deviation.

## Data Availability

All data generated or analyzed during this study are included in this published article (and its Appendix A).

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
