# Peer review of "Nanostructured Layer of Silver for Detection of Small Biomolecules in Surface-Assisted Laser Desorption Ionization Mass Spectrometry"

_materials, 2022, doi:10.3390/ma15124076_

Round 1

Reviewer 1 Report

The authors proposed an inovative methodology for the detection of small molecules based on a "easily" synthesized silver nanostructured layer. The proposed method is described by the authors as accessible for all laboratories without the need of sophisticated equipments, which is one of its interest. The main interest in my opinion being its capacity to be used for the detection of different classes of small molecules.

The paper is well prepared and the methodoly cleraly exposed by the authors.

This paper is, in my opinion, of great interest, and deserves of rapid communication to the community.

I recommend it for publication in Materials.

Author Response

Authors’ response (marked in blue in the attachment) to Reviewers’ comments

Authors are very grateful for critical comments and thoughtful suggestions. Based on these comments and suggestions, we have made a careful revision of the original manuscript. A revised manuscript has been submitted, in which the modified sections are highlighted in red. Thank the Editor and Reviewers again, who made great contributions to improve our paper. Authors believe that the modified text will satisfy the Editor and Reviewers and are ready to answer additional questions.

Editor and Reviewer Comments:

Reviewer 1

The authors proposed an inovative methodology for the detection of small molecules based on a "easily" synthesized silver nanostructured layer. The proposed method is described by the authors as accessible for all laboratories without the need of sophisticated equipments, which is one of its interest. The main interest in my opinion being its capacity to be used for the detection of different classes of small molecules. 

The paper is well prepared and the methodoly cleraly exposed by the authors.

This paper is, in my opinion, of great interest, and deserves of rapid communication to the community.

I recommend it for publication in Materials.

Response: Authors are grateful for appreciation of our manuscript and dedication of time to review our manuscript.

Reviewer 2 Report

Reviewers' comments:

Manuscript ID: materials-1720120

Full Title: Nanostructured layer of silver in surface-assisted laser desorption ionization mass spectrometry for detection of small biomolecules.

The manuscript describes the Nanostructured layer of silver in surface-assisted laser desorption ionization mass spectrometry for detection of small biomolecules. The manuscript needs a detailed editing. 

To improve the quality of this manuscript, I see that the authors must meet the following remarks:

The authors need to consider the following comments

- In the Abstract, the authors need to improve with more specific short results and conclusions, i.e. academic novelty or technical advantages.

- Keywords: keywords should short.

- Introduction - should be improved; more related papers must be discussed and superiority, novelty, critical improvement in this study must be clarified.

- Give more detail for the 2.5. SEM analysis.

- The authors are obliged to repeat the discussion part of the 3.2. SEM analysis.

- In part SEM: how the energy of the accelerator beam used?

- 3.3. SALDI MS performance - section should be detailed.

- References: there are recent references in 2021-2022 treating the same subject, you can use. - And make all references in same format for volume number, page number and journal name.

- Some sentences need reconstruction and the level of English should be improved.

So that I recommended this manuscript to major revision and for future process.

Author Response

Authors’ response (marked in blue in the attachment) to Reviewers’ comments

Authors are very grateful for critical comments and thoughtful suggestions. Based on these comments and suggestions, we have made a careful revision of the original manuscript. A revised manuscript has been submitted, in which the modified sections are highlighted in red. Thank the Editor and Reviewers again, who made great contributions to improve our paper. Authors believe that the modified text will satisfy the Editor and Reviewers and are ready to answer additional questions.

Editor and Reviewer Comments:

Reviewer 2

The manuscript describes the Nanostructured layer of silver in surface-assisted laser desorption ionization mass spectrometry for detection of small biomolecules. The manuscript needs a detailed editing. 

To improve the quality of this manuscript, I see that the authors must meet the following remarks:

The authors need to consider the following comments

- In the Abstract, the authors need to improve with more specific short results and conclusions, i.e. academic novelty or technical advantages.

Response: Authors are grateful for this remark. The abstract has been revised following the Reviewer remark.

- Keywords: keywords should short.

Response: The keywords were shortened following the Reviewer remark.

- Introduction - should be improved; more related papers must be discussed and superiority, novelty, critical improvement in this study must be clarified.

Response: The introduction has been rewritten following the Reviewer remark.

- Give more detail for the 2.5. SEM analysis.

Response: The section ‘2.5.SEM analysis’ has been updated with additional details on SEM analysis following the Reviewer remark.

- The authors are obliged to repeat the discussion part of the 3.2. SEM analysis.

Response: The discussion in the section ‘3.2.SEM analysis’ has been rewritten following the Reviewer remark.

- In part SEM: how the energy of the accelerator beam used?

Response: SEM images were obtained using SEM/FIB instrument (Quanta 3D FEG FEI, Gräfelfing, Germany) operating at 20 kV and in SE mode. This information has been added to the text following the Reviewer remark.

- 3.3. SALDI MS performance - section should be detailed.

Response: Authors are grateful for this remark. The section ‘SALDI MS performance’ has been revised with more detailed discussion.

- References: there are recent references in 2021-2022 treating the same subject, you can use. - And make all references in same format for volume number, page number and journal name.

Response: The recent references were added to the text and introduction, results & discussion sections were revised following the Reviewer recommendation. The references style has been checked and all the mistakes were corrected. Authors are grateful for this comment.

- Some sentences need reconstruction and the level of English should be improved.

Response: The manuscript text has been revised and many sentences were rewritten.

So that I recommended this manuscript to major revision and for future process.

Reviewer 3 Report

This study is about the preparation of a suitable plate for the detection of small biomolecules in surface-assisted laser desorption ionization mass spectrometry. It is interesting although the results are preliminary and could be improved.

Since there are not so many articles like this MS, it can be a good contribution to the field. However, I have some minor comments:

I think the title of MS should be much better and clear when revised as “Nanostructured layer of silver for detection of small biomolecules in surface-assisted laser desorption ionization mass spectrometry”

 Line 51: Do not give only the review article as a reference. Please provide the relevant literature.

Line 168: Please use “instrument” instead of apparatus.

Line 171: Please use “synthesis time” instead of the time of synthesis.

Line 188: Please revise this sentence and make it clear.

Line 191,195, 202, and others: Provide V (unit) for all values.

Line 225: Please avoid using “we” sentences and use passive instead.

Line 308: Here, the discussion is weak and only the results of the other studies were given and compared. It should be better to discuss the reasons and give ideas about the future works to make it better. Please also discuss the RSD values. What might be the reason for high values?

Line 311: Avoid using “we” sentences.

 Please revise the caption of Fig.S1. as SEM images of the additional electrode (stainless-steel plate) at different magnifications.

Please revise the caption of Fig.S3. as “…linear regression equations” instead of “..linear regression..”

It should be better to indicate m/z values for the specific ones as in the case of Fig.2. Otherwise, the researchers cannot understand the MS spectrum.

Author Response

Authors’ response (marked in blue in the attachment) to Reviewers’ comments

Authors are very grateful for critical comments and thoughtful suggestions. Based on these comments and suggestions, we have made a careful revision of the original manuscript. A revised manuscript has been submitted, in which the modified sections are highlighted in red. Thank the Editor and Reviewers again, who made great contributions to improve our paper. Authors believe that the modified text will satisfy the Editor and Reviewers and are ready to answer additional questions.

Editor and Reviewer Comments:

Reviewer 3

This study is about the preparation of a suitable plate for the detection of small biomolecules in surface-assisted laser desorption ionization mass spectrometry. It is interesting although the results are preliminary and could be improved.

Since there are not so many articles like this MS, it can be a good contribution to the field. However, I have some minor comments:

I think the title of MS should be much better and clear when revised as “Nanostructured layer of silver for detection of small biomolecules in surface-assisted laser desorption ionization mass spectrometry”

Response: Authors are thankful for this comment. Indeed, the title proposed by the Reviewer has better readability. Therefore, authors revised the title of the manuscript following the Reviewer recommendation.

Line 51: Do not give only the review article as a reference. Please provide the relevant literature.

Response: We would like to thank the Reviewer for this comment. The reference mentioned by the Reviewer has been replaced.  

Line 168: Please use “instrument” instead of apparatus.

Response: The word ‘apparatus’ has been replaced with ‘instrument’ following the Reviewer recommendation.

Line 171: Please use “synthesis time” instead of the time of synthesis.

Response: The ‘time of synthesis’ has been replaced with ‘synthesis time’ throughout the manuscript text following the Reviewer recommendation.

Line 188: Please revise this sentence and make it clear.

Response: Authors are grateful for this comment. The indicated sentence has been revised to make it clear for the readers.

Line 191,195, 202, and others: Provide V (unit) for all values.

Response: The units (V) have been added for all the values.

Line 225: Please avoid using “we” sentences and use passive instead.

Response: Authors are grateful for this remark. All such sentences were revised following the Reviewer comment.

Line 308: Here, the discussion is weak and only the results of the other studies were given and compared. It should be better to discuss the reasons and give ideas about the future works to make it better. Please also discuss the RSD values. What might be the reason for high values?

Response: Authors are grateful for this recommendation, which indeed improved the quality of the manuscript. The whole discussion has been rewritten with consideration of LOD and RSD values. The obtained values were discussed in terms of SALDI MS sensitivity and application of analysis of low molecular weight compounds using the target synthesized with the proposed approach. High values of RSD were obtained for concentrations determined as LOD, while for e.g. for concentration of glucose 1 μg/mL the value was 11.08%. 

Line 311: Avoid using “we” sentences.

Response: All the ‘we’ sentences were revised with the use of passive voice.

Please revise the caption of Fig.S1. as SEM images of the additional electrode (stainless-steel plate) at different magnifications.

Response: The caption of Fig.S1 has been revised following the Reviewer recommendation.

Please revise the caption of Fig.S3. as “…linear regression equations” instead of “..linear regression..”

Response: The caption of Fig.S3 has been revised following the Reviewer recommendation.

It should be better to indicate m/z values for the specific ones as in the case of Fig.2. Otherwise, the researchers cannot understand the MS spectrum.

Response: Fig. S2 has been revised following the Reviewer recommendation.

Round 2

Reviewer 2 Report

Reviewers' comments:

The authors revised the manuscript according to the reviewers' comments.